# PartInfer: Enabling LLM Inference On Edge Devices[*]

## Abstract

Large Language Models (LLMs) have demonstrated remarkable capabilities across a range of Natural Language Processing (NLP) tasks, but their high computational and memory demands pose significant challenges for deployment on resource-constrained edge devices. Existing approaches to model compression and optimization often rely on coarse-grained pruning or quantization, which can compromise accuracy or require re-training and fine-tuning. In this work, we introduce **PartInfer**, a neuron-level optimization framework that enables efficient LLM inference on edge devices by exploiting the task-specific activation patterns of neurons. By profiling and identifying both task-specific and general-purpose neurons using an *offline LLM profiler*, PartInfer implements two key optimizations: *partial loading*, which reduces memory footprint by loading only a subset of neurons that were identified to be most important during the offline stage, and *partial computation*, which dynamically computes only the most relevant neurons at runtime. Evaluation across multiple NLP tasks shows that PartInfer achieves significant reductions in memory footprint and computation while preserving task performance, making it a practical step towards enabling LLM deployment on edge devices.

## 1 Introduction

Large Language Models (LLMs) have rapidly become foundational tools across a wide array of applications, including natural language writing, code generation, healthcare diagnostics, financial analysis, and many more (Touvron et al., 2023; Daivi, 2024; Anastasiya Zharovskikh, 2023; Cell-Strat, 2023). Their impressive capabilities have driven their widespread adoption, predominantly through deployment in powerful data centers equipped with high-end GPUs. These centralized environments offer the immense computational resources required to run LLM inference efficiently.

However, growing concerns around user privacy, the need for application customization, and the demand for offline capabilities (Chun et al., 2011; Xue et al., 2024; by PrivateGPT, 2023; Lyu et al., 2023) have fueled increasing interest in enabling LLM inference directly on edge devices, such as Nvidia Jetson chips (NVIDIA Corporation, 2025b) or Google Coral (goo, 2025). Edge deployment offers the potential for greater data security and responsiveness, but it also faces significant challenges. Unlike data centers, edge devices are constrained by limited memory and computational power, which complicates the direct execution of large, resource-hungry models.

Prior research has explored various strategies to mitigate these constraints. Techniques such as quantization (Dettmers et al., 2023; Lin et al., 2024) and compression (Choudhary et al., 2020) reduce model size and computation but often degrade output quality and lack hardware generality, e.g., aggressive quantization schemes like three-bit quantization are not universally supported across GPU architectures (Kim et al., 2021). Pruning methods (Bansal et al., 2022; Ma et al., 2023) modify the model structure by removing less important parameters but usually require costly re-training or fine-tuning to restore performance. Offloading parts of the model to the CPU (Aminabadi et al., 2022; Cai et al., 2023) can alleviate GPU memory bottlenecks, but this approach is limited by data transfer speeds over PCIe, necessitates powerful CPUs, and is incompatible with unified CPU-GPU memory architectures such as those found in NVIDIA Jetson devices (NVIDIA Corporation, 2025a), Google Coral boards (goo, 2025), or even Qualcomm Snapdragon SoCs (Dagli & Belviranli, 2024).

---

[*]LLMs were used for polishing the writing of a preliminary version of the paper.

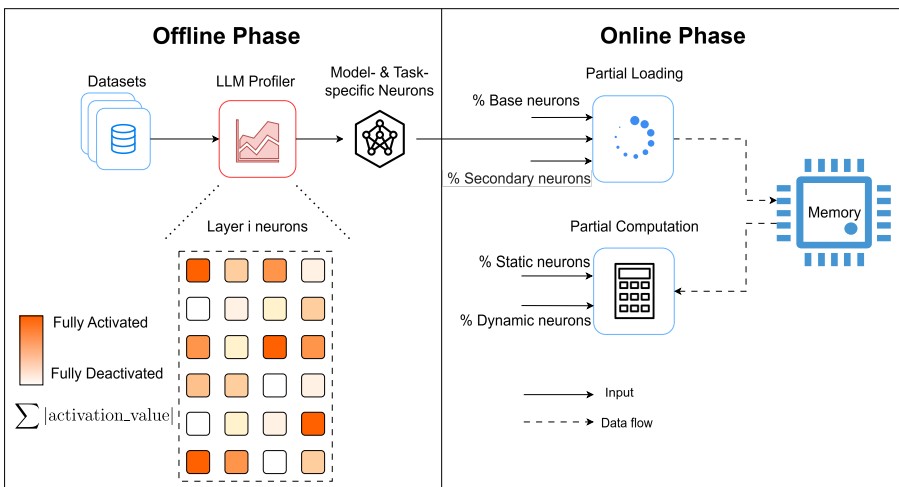

Figure 1: Overview of PartInfer. Our approach has two phases: offline (left) and online (right). In the offline phase, we run LLM Profiler to generate neuron files. In the online phase, partial loading reduces memory usage, while partial computation lowers computational overhead.

An alternative and promising line of work has focused on neuron sparsity during inference. It has been observed that only a small subset of neurons activate and meaningfully contribute to each output, exhibiting a skewed distribution where roughly 10–20% of neurons consistently dominate the model's output quality (Liu et al., 2023; Alizadeh et al., 2024; Mirzadeh et al., 2023). Building on this insight, recent methods like PowerInfer (Song et al., 2024), PowerInfer2 (Xue et al., 2024), and LLM-in-a-Flash (Alizadeh et al., 2024) train lightweight MLP predictors at each model layer to dynamically identify the active neurons at runtime. While effective, these predictors add an additional memory footprint (approximately 15–20%) and impose computational overhead.

Another noteworthy approach, CoreInfer (Wang et al., 2024), claims to identify activated neurons during the prefill phase of inference. CoreInfer demonstrates that these neurons show stability and semantic similarity linked to the input sentence and tend to cluster according to the domain of the input. This clustering suggests a strong correlation between neuron activation patterns and the semantic context of tasks. However, CoreInfer requires the loading of the model completely into the memory and thus does not reduces the memory footprint. Furthermore, it does not exploit model-level knowledge and only selects neurons based on the prefilling stage of the current prompt.

In light of this challenge, our approach, **PartInfer**, extends these insights further by identifying active neurons offline, before runtime. Following CoreInfer's observations, we categorize neurons into two groups: *base neurons*, which activate selectively depending on the input's task ( e.g., QA, translation, or summarization), and *secondary neurons*, which serve general purposes and activate broadly across inputs. Our approach is summarized in Figure 1. As shown on the left side, using an offline **LLM profiler**, we accurately determine these neuron groups. To address memory constraints, in the online phase (right side of Figure 1), we implement **partial loading**, a mechanism that loads only the relevant neurons identified by the offline profiler into memory, discarding the remaining ones. Additionally, we introduce **partial computation**, which selectively computes only the most relevant neurons during runtime, further accelerating inference.

We validate PartInfer across three lightweight LLMs, Llama3.2-3B (Meta AI, 2023b), Llama3.2-1B (Meta AI, 2023a), and Qwen2.5-3B (Team, 2024), on the NVIDIA Jetson Orin Nano (NVIDIA Corporation, 2025b), a highly memory-constrained device with 8 GB of shared CPU–GPU memory. Without our approach, the 3B-scale models cannot be executed due to their 8.5–9.5 GB memory requirements including framework overhead, leaving only ~6.5 GB available on the device. With *partial loading*, PartInfer enables both 3B-scale models to run by loading only the most relevant neurons identified during offline profiling. Across QA, translation, and summarization tasks, PartInfer preserves accuracy while yielding substantial speedups; for example, Llama3.2-3B reaches $\approx 11$ tokens/s, achieving more than a $13\times$ improvement over disk-offloading baselines. Similar gains are observed for Llama3.2-1B and Qwen2.5-3B, demonstrating that PartInfer generalizes across model families and offers a practical path to deploy LLMs on resource-constrained edge hardware.

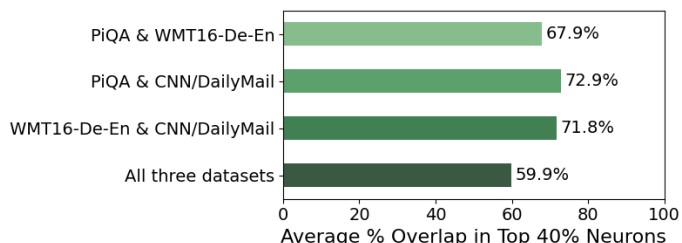

Figure 2: High intra-task neurons overlap. This proves that there is a set of neurons that always get activated, regardless of the input task.

In summary, our contributions include an offline LLM profiler that accurately identifies model-wide and task-specific neurons, a novel partial loading scheme that enables deployment of models beyond conventional memory limits on edge devices, and a partial computation mechanism that selectively processes relevant neurons to accelerate inference without sacrificing accuracy.

## 2 KEY INSIGHTS

This section highlights the key insights that shaped the development of our approach.

### 2.1 SPARSITY IN LLMS

A key property of LLMs is their inherent *sparsity*: only a small subset of neurons is strongly activated for a given input, largely determining output quality (Liu et al., 2023; Alizadeh et al., 2024; Mirzadeh et al., 2023). Thus, the model's full capacity is rarely used at once; instead, different parts become selectively active depending on the task or input distribution. Building on this, we further analyze neuron roles across tasks, distinguishing task-specific from model-specific neurons.

### 2.2 MODEL-SPECIFIC NEURONS

We define *model-specific neurons* as neurons that are consistently activated across tasks, regardless of input type. To quantify this, we aggregated prompts from three distinct datasets, namely PiQA (Bisk et al., 2020b) for question answering, CNN/DailyMail (See et al., 2017; Hermann et al., 2015a) for summarization, and WMT16 DE-EN (Bojar et al., 2016) for translation, and measured neuron activations across 100 prompts per task using Llama3.2-3B (Meta AI, 2023b) model. We then selected the top 40% of neurons (ranked by absolute activation value) for each layer and examined their overlap across tasks. The results are averaged across layers. We refer to Section 6 for more details on the evaluation setup.

The results are shown in Figure 2. Each bar represents the overlap of the top 40% neurons for each task pair. We observe a significant cross-task overlap of approximately 68% to 72% within the top 40% of neurons, and approximately 60% overlap between all three datasets. This provides strong evidence that a stable core of neurons is universally engaged across tasks. These neurons represent general-purpose computation in the model and may correspond to linguistic or representational features that are task-agnostic.

### 2.3 TASK-SPECIFIC NEURONS

In addition to general-purpose neurons, we find strong evidence for *task-specific neurons*. These are neurons that consistently activate within a given task but differ across tasks. To quantify this, for each dataset, we create ten chunks of ten prompts each, identified the top 40% most active neurons per chunk, and measured the average overlap among all chunks. The results are averaged across all layers. As shown in Figure 3, the intra-task overlap ranges from 58.6% for PiQA to 69.5% for CNN/DailyMail. This indicates that for each task, a substantial subset of neurons is consistently responsible for processing task-specific information. Furthermore, to confirm that these subsets are distinct across tasks, we picked these top 40% neurons for each task, and calculated the overlap

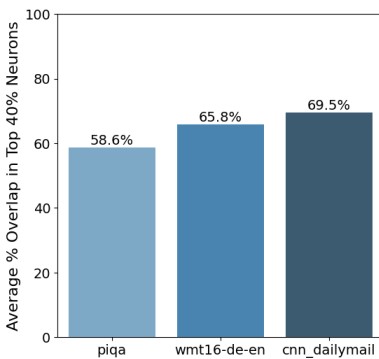
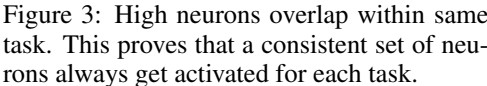

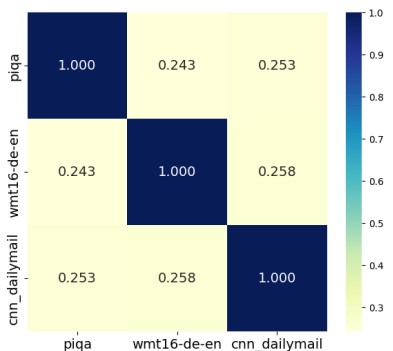

Figure 3: High neurons overlap within same task. This proves that a consistent set of neurons always get activated for each task.

Figure 4: Low inter-task neurons overlap. This proves that each task activates its own set of neurons.

across tasks. As shown in Figure 4, the cross-task overlap is only around 24 to 26% which is far lower than intra-task overlap. This highlights that although each task reuses some task-specific neurons, it also recruits its own specialized subset of neurons.

These results show a dual structure: a set of model-specific neurons that are universally engaged across tasks, and a set of task-specific neurons that drive performance for individual tasks. This interplay between generality and specialization is a key property of LLM computation in PartInfer.

## 3  LLM PROFILER

In the offline profiling stage, we run selected datasets through the model to measure neuron activations. In Transformer Feed Forward Network (FFN) blocks (e.g., `gate_proj`, `up_proj`, `down_proj` in Llama3 and Qwen2.5), each neuron corresponds to a column in the gate/up projections and a row in the down projection. During profiling, we record the absolute activations at the input of the `down_proj` layer (i.e., the output of the activation function), which directly correspond to these FFN neurons. For each prompt, we record the absolute values of neuron activations, along with the number of input and generated tokens. Since activation functions vary across architectures, the scoring method must sometimes be adapted. For models using SwigLU, we define the following scoring function: Let $\mathcal{T}$ be the set of tokens processed during scoring and let $a_n(t)$ denote the activation of neuron $n$ on token $t \in \mathcal{T}$. Then, for each neuron $n \in \{1, \ldots, N\}$, we define $s_n$ as the score of the neuron:

$$s_n = \sum_{t \in \mathcal{T}} |a_n(t)| . \tag{1}$$

The scores of all neurons, along with the token counts, are stored in score files. These files can be merged in different configurations by normalizing scores by token count and summing the normalized values. Task-specific neurons can be obtained by merging score files from the same task, while model-level neurons can be obtained by merging across tasks. The final output is a file containing neuron indices ranked in decreasing order of importance across the evaluated datasets. We later discuss the memory and computation cost of this profiler in Appendix B

## 4  PARTIAL LOADING

Following the analysis conducted by the LLM Profiler, the extracted insights are used to optimize the deployment of LLMs on edge devices by reducing their computational and memory requirements. We first focus on minimizing the memory footprint, which allows larger models to be deployed on resource-constrained devices where full model loading is otherwise infeasible.

The core idea is to partially load the model into memory. Partial loading is applied to the FFN as this component accounts for the largest proportion of the total model parameters. To reduce the memory usage of the FFN, we selectively load only a subset of its neurons, focusing on those considered most

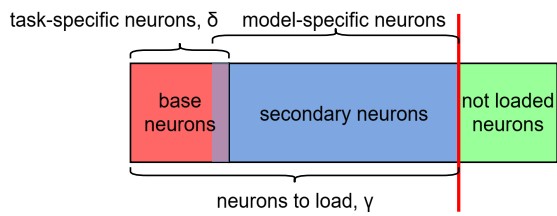

Figure 5: Selection of neurons to load during partial loading.

relevant. This means that, for every block in the FFN (e.g., `gate_proj`, `up_proj`, `down_proj` in Llama3 or Qwen2.5), we select the most important neurons, and load the corresponding slices to the memory, and discard the rest. The remaining components of the model are fully loaded. This selective loading is guided by the neuron relevance profiled by the LLM Profiler during the offline phase. Our method classifies FFN neurons into two groups, *base neurons* and *secondary neurons*. As illustrated in Figure 5, we first select the top $\delta$ most relevant **task-specific** neurons to form the base neurons, and then add the most relevant **model-specific** neurons as secondary neurons until the overall loading threshold $\gamma$ is reached.

Based on memory constraints and application requirements, we specify the proportion of neurons to be loaded. This selection is applied independently to each FFN layer in the model. When loading the model and reading neurons from disk, only the weights corresponding to these selected neurons are loaded into memory, leading to substantial memory savings while preserving model performance.

## 5 PARTIAL COMPUTATION

While partial loading effectively alleviates memory constraints by keeping only a subset of neurons in memory, the computational cost of inference remains significant. To address this, we propose *partial computation*, a runtime mechanism that selectively computes only the most relevant **loaded** neurons. This strategy reduces computational overhead and accelerates inference without significantly degrading output quality.

As shown in our earlier analysis in Section 2, a relatively small fraction of neurons is sufficient to preserve model quality if they are chosen judiciously. For example, even though $\gamma$ of neurons may be loaded into memory due to partial loading, we can further reduce computation by only evaluating $\epsilon$ of neurons at runtime ($\epsilon < \gamma$). This trade-off between accuracy and speed is user-configurable, enabling flexible deployment depending on application requirements. During computation, we classify neurons into two categories: *static neurons* and *dynamic neurons*. **Static neurons** are identical to the base neurons selected during partial loading. These task-specific neurons are always computed for every prompt, regardless of the input. **Dynamic neurons** are selected at runtime based on the input prompt. Their selection is adaptive and varies per prompt, allowing the computation process to focus on neurons that are most relevant for the current context.

Figure 6 illustrates the relationship between static and dynamic neurons in the context of partial computation. The selection of dynamic neurons occurs during the *prefill phase* of inference, inspired by CoreInfer's (Wang et al., 2024) methodology. Figure 7 illustrates this process. For each token, we compute activation values for all loaded neurons. We sort these activations in descending order and select the top $\alpha$ neurons for that token and record the indices of these top $\alpha$ neurons. This is repeated for all tokens in the prompt, resulting in multiple lists of top neurons. From these lists, we count the frequency of neuron occurrences and select the top $\beta$ neurons with the highest frequency. These $\beta$ neurons form the dynamic set for this prompt. In the end, the static neurons are then combined with the selected dynamic neurons until the total computation ratio $\phi = \delta + \epsilon$ is reached. This approach ensures that computation resources are concentrated on neurons most likely to contribute to the output for the given input, achieving substantial speedups while retaining accuracy.

## 6 EVALUATION

In this section, we evaluate PartInfer across three key performance dimensions that are crucial for deploying LLM on memory- and compute-constrained edge devices: **accuracy**, **memory footprint**,

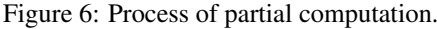

Figure 6: Process of partial computation.

Figure 7: Dynamic neuron selection process.

and **decoding speed**. Prior approaches typically optimize only one or two of these metrics. For example, 4-bit quantization excels in memory and accuracy but suffers from slower decoding and limited flexibility, whereas CoreInfer achieves high decoding speed but offers neither strong accuracy nor memory reduction. PartInfer's goal is to develop an approach that balances all three. It exposes configurable knobs that allow users to tune the accuracy–memory–speed operating point depending on device constraints or application requirements. We now describe our experimental setup and evaluate these trade-offs in detail.

### 6.1 SETUP

For our evaluation, we use the NVIDIA Jetson Orin Nano Developer Kit (NVIDIA Corporation, 2025a), which integrates a 6-core Arm Cortex-A78AE CPU, an NVIDIA Ampere GPU with 1,024 CUDA cores and 32 Tensor Cores, and 8 GB of LPDDR5 shared memory. We benchmark three representative models, Llama3.2-3B, Llama3.2-1B, and Qwen2.5-3B, to compare PartInfer against two strong baselines: CoreInfer and 4-bit bitsandbytes quantization.

### 6.2 DATASETS AND NEURON-FILES SELECTION

In our setup, we evaluate three task categories (question answering, translation, and summarization) using multiple datasets within each domain. For question answering, we use SQuAD (Rajpurkar et al., 2016), TriviaQA (Joshi et al., 2017), MLQA (Lewis et al., 2019), and PIQA (Bisk et al., 2020a). For translation, we use WMT16-de-en and WMT16-en-de (Bojar et al., 2016), WMT14-fr-en and WMT14-en-fr (Bojar et al., 2014). For summarization, we use CNN/DailyMail (Hermann et al., 2015b), Samsum (Gliwa et al., 2019), and XSum (Narayan et al., 2018).

Task-specific neuron files are generated in the offline stage using all datasets within a task, ensuring that the resulting neurons reflected consistent task-specific responses. In contrast, model-specific neurons are derived by applying the same scoring method to a cross-task mixture constructed from one dataset per task, identifying neurons that generalize beyond a single task. Specifically, we use PIQA, WMT14-en-fr, and Xsum datasets to generate the model-specific neuron file. We observed that neuron scores stabilize after only a few datasets, indicating that the number of datasets is sufficient for reliable results. Moreover, the specific choice of datasets for model-specific neuron calculation has little impact on scores, as long as they were drawn from different task groups.

### 6.3 PARAMETER CONFIGURATION

We found empirically that static neurons should correspond to task-specific neurons, while dynamic neurons represent model-specific behavior. Since task-specific neurons are more directly tied to the target objective, they exhibit higher stability across prompts. For instance, with $\delta = 0.3$, Figure 3 shows that roughly $60\%$ of task-specific neurons overlap across prompts within the same task.

PartInfer exposes several tunable parameters (see Table 1). Their values depend on model size, device constraints (e.g., memory capacity), expected speedup, and the desired accuracy. In practice, hardware limitations largely determine the feasible percentage of neurons that can be loaded, while the allocation between base and secondary neurons and the fraction of computed neurons must be empirically validated. For the loading ratio $\gamma$, we observed that $\gamma = 0.7$ is the maximum value at which the 3B-scale models fits on the NVIDIA Jetson Orin Nano 8GB without running

Table 1: Important parameter values used in our experiments for all three models.

| Parameter | Value |
|---|---|
| First layer | 5 |
| Last layer | -2 (Last 2 are skipped) |
| Percentage of base / static neurons | 30% |
| Percentage of dynamic neurons | 10% |
| Percentage of computed neurons (sparsity) | 40% |
| Percentage of secondary neurons | 40% |
| Percentage of overall loaded neurons | 70% |

Table 2: Benchmark results for PartInfer and CoreInfer variants across three models and six datasets.

| Model | Method (Metric) | SQuADv2 (Exact Match) | MLQA en–en (Exact Match) | WMT16 de–en (BLEU) | WMT16 ro–en (BLEU) | XSUM (Rouge) | CNN/DailyMail (Rouge) |
|---|---|---|---|---|---|---|---|
| **Llama3.2-1B** | PartInfer | **13.75** | **0.40** | **10.50** | **10.85** | **0.12** | **0.15** |
| | CoreInfer | 6.00 | 0.35 | 2.91 | 2.87 | 0.08 | 0.13 |
| | CoreInfer + Partial Loading | 3.85 | 0.28 | 1.68 | 1.39 | 0.05 | 0.11 |
| | CoreInfer + Random Loading | 1.26 | 0.00 | 0.03 | 0.03 | 0.04 | 0.03 |
| **Llama3.2-3B** | PartInfer | **22.55** | 0.52 | **25.68** | **25.96** | **0.12** | **0.17** |
| | CoreInfer | 17.37 | **0.53** | 4.97 | 4.98 | 0.00 | 0.15 |
| | CoreInfer + Partial Loading | 11.60 | 0.44 | 1.70 | 1.65 | 0.10 | 0.12 |
| | CoreInfer + Random Loading | 0.26 | 0.01 | 0.04 | 0.03 | 0.00 | 0.05 |
| **Qwen2.5-3B** | PartInfer | 24.32 | **0.53** | **8.28** | **4.83** | 0.10 | 0.15 |
| | CoreInfer | 32.58 | 0.53 | 4.32 | 4.09 | **0.11** | **0.17** |
| | CoreInfer + Partial Loading | **35.69** | 0.48 | 1.55 | 0.80 | 0.10 | 0.16 |
| | CoreInfer + Random Loading | 29.65 | 0.34 | 0.44 | 0.16 | 0.07 | 0.13 |

out of memory. To ensure consistent comparison across model sizes, we apply the same value to Llama3.2-1B. The percentage of computed neurons $\phi$, follows prior analyses from CoreInfer, which shows that computing the top $40\%$ of neurons provides a strong accuracy–efficiency trade-off. Our experiments confirmed that this setting works well across tasks and models. Moreover, we have empirically proven our selection of the layer range and the division between base and dynamic neurons. More detailed analysis is provided in Appendix A. Overall, the parameter values presented in Table 1 are chosen based on hardware feasibility, prior validated methodology, and targeted empirical exploration rather than arbitrary selection.

### 6.4 ACCURACY

Table 2 summarizes the accuracy of our proposed **PartInfer** method compared to three baselines: the original CoreInfer (Wang et al., 2024), an enhanced version incorporating our partial loading (CoreInfer + Partial Loading), and a control condition where neurons are loaded at random (Core-Infer + Random Loading). We evaluate these methods on six datasets across three model families: Llama3.2-1B, Llama3.2-3B, and Qwen2.5-3B. The results collectively highlight the complementary benefits of PartInfer, and demonstrate that its advantages generalize across model families.

**Overall superiority of PartInfer.** Across all models and nearly all datasets, PartInfer achieves the highest accuracy. For Llama3.2-3B and Llama3.2-1B, PartInfer consistently reaches substantially higher scores than CoreInfer, particularly on the translation tasks WMT16 DE–EN and RO–EN. On the 3B model, PartInfer attains scores around 25–26 BLEU, compared to only 4–5 BLEU for CoreInfer. On the 1B model, PartInfer reaches roughly 10–11 BLEU, again far above CoreInfer's 2–3 BLEU range. For Qwen2.5-3B model, the results further confirm this trend. PartInfer achieves the strongest performance on both translation tasks (8.28 BLEU on DE–EN and 4.83 BLEU on RO–EN), and remains competitive on MLQA en–en, and both summarization tasks, where it matches or closely trails CoreInfer. The only exception is SQuADv2 dataset, where PartInfer achieves less accuracy (24.32) than CoreInfer (32.58). Notably, CoreInfer + Partial Loading (35.69) outperforms CoreInfer, eventhough it loads only 70% of neurons. Another example is in Llama3.2B model in Xsum dataset, where CoreInfer + Partial Loading, achieves 0.1, while CoreInfer achieves 0. We attribute this to the offline profiling stage acting as a beneficial pre-filter: by loading only neurons consistently activated across the profiling tasks, some task-irrelevant or noisy neurons may be removed, occasionally helping CoreInfer focus on a more relevant subset.

Table 3: Benchmark results using Dense and Quantization methods. For dense, we load 70% of neurons and compute all of them. For quantization, we use 4-bit bitsandbytes (Bitsandbytes, 2025).

| Model | Method (Metric) | SQuADv2 (Exact Match) | MLQA en–en (Exact Match) | WMT16 de–en (BLEU) | WMT16 ro–en (BLEU) | XSUM (Rouge) | CNN/DailyMail (Rouge) |
|---|---|---|---|---|---|---|---|
| **Llama3.2-1B** | 70% Dense | **17.87** | **0.44** | **27.34** | **25.95** | 0.14 | 0.16 |
| | 4-bit Quantized | 13.97 | 0.43 | 26.70 | 24.65 | **0.15** | **0.17** |
| **Llama3.2-3B** | 70% Dense | 26.41 | 0.55 | 34.91 | 33.40 | 0.21 | 0.18 |
| | 4-bit Quantized | **30.14** | **0.59** | **37.64** | **35.03** | **0.22** | **0.19** |
| **Qwen2.5-3B** | 70% Dense | 31.06 | 0.58 | **27.97** | **20.29** | **0.14** | **0.18** |
| | 4-bit Quantized | **34.32** | **0.59** | 18.18 | 11.30 | 0.12 | 0.17 |

**Effectiveness of partial loading.** Comparing CoreInfer + Partial Loading against CoreInfer + Random Loading isolates the benefit of profiling-based neuron selection. Random loading severely damages performance in every setting, often collapsing accuracy to near-zero values across all six tasks. In contrast, partial loading maintains a meaningful proportion of CoreInfer's performance, even though only $\gamma = 70\%$ of neurons are available. This stability holds across all evaluated models. On Qwen2.5-3B, for example, CoreInfer + Partial Loading achieves BLEU scores of $1.55$ and $0.80$ on the translation tasks, whereas random loading yields only $0.44$ and $0.16$. Although partial loading naturally reduces performance relative to full-neuron CoreInfer, it clearly preserves more accuracy than random selection. This demonstrates that profiling is essential for identifying consistently active and task-relevant neurons.

**Contribution of partial computation.** To understand the effect of partial computation itself, we compare PartInfer with CoreInfer + Partial Loading, which shares the same set of loaded neurons but differs during computation. Across all models and nearly all datasets, PartInfer consistently and substantially outperforms its partial-loading-only counterpart. The gaps are particularly large on the translation datasets: on Llama3.2-3B, PartInfer achieves 25.68 BLEU on WMT16 DE–EN compared to only 1.70 for CoreInfer + Partial Loading, and similar patterns appear on Llama3.2-1B and Qwen2.5-3B. These results demonstrate that profiling-guided partial computation is crucial for identifying and prioritizing the most informative neurons during inference.

**Comparison with dense and quantized baselines.** Table 3 extends our analysis by evaluating an additional baseline, **4-bit quantized** variant using bitsandbytes (Bitsandbytes, 2025) quantization, which is commonly used to reduce inference cost and memory footprint. To make it a fair comparison, we compare it against 70% dense configuration, where only 70% of neurons are loaded using our partial loading, but all of them are fully computed. Across all three model families, 4-bit quantization is competitive with or superior to the 70% dense baseline. Across all three models, the 70% dense baseline proves to be a strong and reliable reference point, often outperforming or matching the 4-bit quantized version, especially on translation datasets. For Llama3.2-1B, the dense baseline dominates on WMT16 DE–EN and RO–EN, reaching 27.34 and 25.95 BLEU, whereas the quantized version achieves slightly lower values (26.70 and 24.65). This pattern persists in the Qwen2.5-3B model, where dense loading achieves substantially stronger translation performance (27.97 BLEU on DE–EN and 20.29 on RO–EN), while quantization drops to 18.18 and 11.30 BLEU. On the other hand, the quantized version performs competitively on QA tasks (e.g., MLQA and SQuADv2), and occasionally surpasses the dense version on Llama3.2-3B and Qwen2.5-3B. However, even in these settings, the differences are generally small, and the dense baseline remains competitive and stable across all tasks.

**Summary.** These results validate the dual contribution of our mechanisms: partial loading improves memory efficiency without degrading performance, while partial computation boosts performance by focusing on the most relevant neurons. Additionally memory footprint and speedup can be controlled independently. Available memory can be fully utilized while speedup can be aligned to available computational resources and application requirements.

## 6.5 Memory Footprint

A key motivation behind PartInfer is reducing the memory required to load and execute LLM on resource-constrained edge devices. Unlike full-parameter loading approaches, PartInfer stores only

Table 4: End-to-end memory usage (in GB) for different inference approaches across three models.

| Model | Original | PartInfer | CoreInfer | Quantized (4-bit) |
|-------|----------|-----------|-----------|-------------------|
| Llama3.2-3B | 6.88 | 5.97 | 7.44 | 3.03 |
| Llama3.2-1B | 2.86 | 2.60 | 2.86 | 1.52 |
| Qwen2.5-3B | 6.48 | 5.37 | 7.52 | 2.67 |

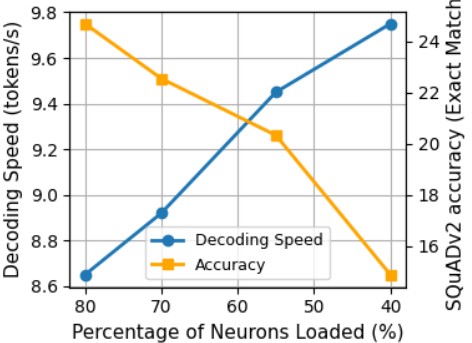

Figure 8: Effect of partial loading on decoding speed and accuracy using Llama3.2-3B.

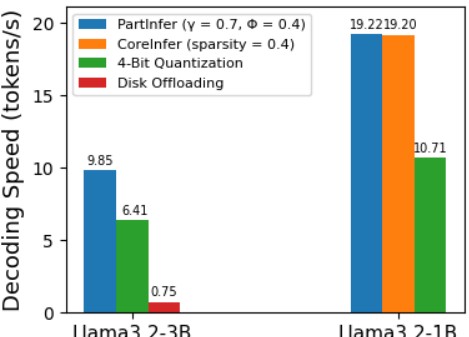

Figure 9: Comparison of throughput between different approaches.

a subset of neurons in memory. The theoretical reduction in memory usage can be expressed as

$$\Delta M = N_{\text{ffn}} \times (1 - r_{load}) \times r_{layer} \times s \qquad (2)$$

where $N_{\text{ffn}}$ denotes the number of FFN parameters, $r_{load}$ is the fraction of neurons preloaded into memory, $r_{layer}$ is the fraction of layers where PartInfer is applied, and $s$ is the storage size per parameter. For example, in Llama3.2-3B, where $N_{\text{ffn}} \approx 2.1B$, choosing $r_{load} = 0.7$, $r_{layer} = 0.75$, and $s = 2$ bytes (`float16`) yields $0.945$ GB, aligning with the value observed during deployment.

To contextualize these analytical expectations, we further report the end-to-end memory consumption of PartInfer against three baselines: the original model (full-parameter loading), CoreInfer, and 4-bit bitsandbytes (Bitsandbytes, 2025) quantization. Table 4 summarizes the peak memory usage across three models: Llama3.2-3B, Llama3.2-1B, and Qwen2.5-3B.

As expected, the original model incurs the full memory footprint because all neurons must be loaded in memory during inference. CoreInfer consistently exceeds the reference memory usage due to additional buffers and intermediate structures needed to support its core neurons computation. In contrast, PartInfer achieves a lower memory footprint across all evaluated models because it loads only a fraction of neurons into memory (70% in our experiments).

The reduction is most noticeable in the two 3B-scale models. For Llama3.2-3B, PartInfer reduces memory from 6.88 GB to 5.97 GB, corresponding to a savings of $\sim 13\%$, consistent with the analytical estimate in equation 2. A similar trend appears in Qwen2.5-3B, where PartInfer achieves a reduction of over 1.1 GB relative to the reference model. For the 1B-scale model, the improvement is smaller: from 2.86 GB to 2.6 GB. This is because PartInfer is applied to fewer layers in Llama3.2-1B (9 of 16 layers), reducing the relative impact of partial loading compared to larger models where a higher proportion of layers are covered. Finally, 4-bit quantization unsurprisingly yields the lowest memory footprint overall, as it compresses all weights, whereas PartInfer selectively loads fewer neurons but retains full precision for the loaded parameters. However, quantization trades off memory reduction for lower decoding throughput, as we will discuss in the next subsection.

## 6.6 SPEEDUP

In this section, we evaluate the impact of PartInfer on decoding throughput (tokens/s) and task accuracy using Llama3.2-3B and Llama3.2-1B models. The decoding speed corresponding to varying proportions of loaded neurons is illustrated in Figure 8. Reducing the fraction of loaded neurons

leads to an increase in prefilling and decoding stage, since fewer weights are fetched from memory and computed. In our example, loading only $\gamma = 40\%$ of the neurons improves throughput from 8.65 token/s to 9.75 tokens/s compared to 80% loading. However, this comes at the cost of accuracy: on SQuADv2, accuracy improves from 14.87 at 40% loaded neurons to 24.70 at 80%.

**Comparison with other baselines**. We evaluate the decoding performance of our proposed approach, PartInfer, against three baselines on the NVIDIA Jetson Orin Nano platform: CoreInfer, 4-bit bitsandbytes (Bitsandbytes, 2025) quantization, and disk offloading. Figure 9 summarizes these results. For the Llama3.2-3B model, PartInfer attains 9.85 tokens/s, outperforming 4-bit quantization (6.41 tokens/s) by roughly $1.5\times$ and exceeding disk offloading (0.75 tokens/s) by more than $13\times$, the latter being heavily bottlenecked by frequent parameter transfers between disk and GPU memory. CoreInfer is excluded from the 3B comparison because it cannot fit within the memory constraints of the device. The substantial speedup of PartInfer arises from selectively computing only 40% of neurons at runtime, in contrast to quantized inference, which evaluates all neurons. For the Llama3.2-1B model, PartInfer and CoreInfer achieve comparable performance ($\sim$19.2 tokens/s), as both similarly restrict computation to 40% of neurons. Meanwhile, the quantized model reaches only 10.71 tokens/s, approximately half the speed of the previous approaches.

## 6.7 SUMMARY

Our evaluation shows that PartInfer provides a balanced and tunable trade-off across accuracy, memory usage, and decoding speed, three metrics that existing approaches struggle to optimize simultaneously. Although 4-bit quantization achieves the smallest memory footprint (about half of PartInfer's; Table 4), PartInfer reaches comparable accuracy while delivering **1.5–2$\times$** higher decoding throughput (Figure 9), and it avoids the hardware limitations of low-precision kernels, which are not consistently supported across GPUs (NVIDIA Corporation, 2025c; Jamie Dborin, 2025). Moreover, quantization offers only coarse control (typically 4-bit vs. 8-bit), making it difficult to fully utilize the available memory or computation of an edge device. In contrast, PartInfer exposes fine-grained knobs that allow users to adjust both the percentage of loaded and computed neurons, enabling precise memory–speed–accuracy trade-offs and allowing the model to scale up or down to match device capabilities. Compared to CoreInfer, with fast decoding speed but less accurate and without memory savings, PartInfer provides a more robust and adaptable inference framework.

## 7 CONCLUSION

This paper presents PartInfer, a neuron-level optimization framework for deploying LLM on edge devices. It provides fine-grained control over memory footprint and computational requirements independently while maintaining high accuracy of the generated outputs. We define model-specific and task-specific neurons highlighting their distinct roles. We exploit this insight to introduce partial loading to reduce the memory footprint and partial computation to lower computational requirements. These optimizations are realized by model knowledge extracted during an offline analysis phase. Evaluations demonstrate effectiveness in enabling LLM inference on resource-constrained devices. Experimental results show the effectiveness of PartInfer on memory footprint and speedup while maintaining high acccuracy.

## 8 FUTURE WORK

Future work will extend the current approach beyond CPU and GPU execution by incorporating support for neural processing units (NPUs), such as those available on Jetson devices. In addition to hardware adaptation, further research will address the current limitation of manually identifying task-specific neurons offline. This could be addressed by integrating lightweight classifiers that allow dynamic neuron selection during inference based on the task type. Finally, to broaden the applicability of the approach, future efforts will focus on generalizing the method to a wider range of models and tasks.

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

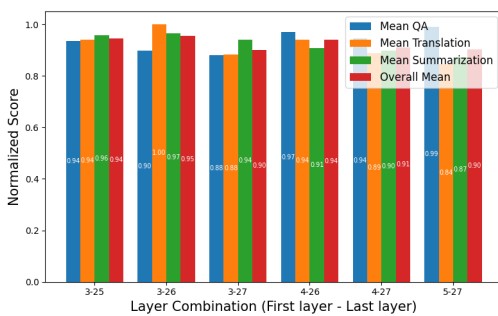
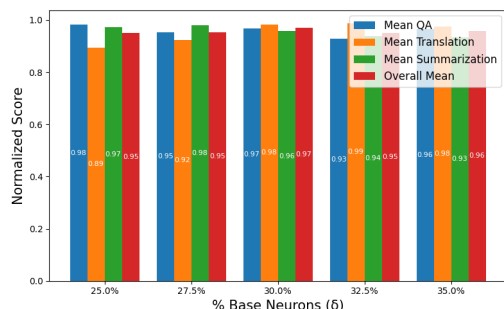

Figure 10: Impact of different layer combinations on cross-task performance. The values are normalized to the maximum value of each task.

Figure 11: Effect of $\delta$ on performance across different tasks. The values are normalized to the maximum value of each task.

Table 5: Offline profiler cost per model and number of processed tokens.

| Model | Tokens processed (M) | Time (h) | Throughput (tokens/s) | Peak GPU memory (GB) |
|---|---|---|---|---|
| Qwen2.5-3B | 43.9 | 5.51 | 2214 | 16.0 |
| Llama3.2-3B | 43.7 | 5.01 | 2426 | 21.8 |
| Llama3.2-1B | 8.8 | 0.56 | 4349 | 7.6 |
| Llama3.2-1B | 43.8 | 2.69 | 4518 | 10.8 |

## A    ABLATION STUDY

In this section, we explain the chosen parameter values shown in Table 1.

**Layer-range selection**. We evaluated multiple first/last layer combinations using the Llama3.2-3B model (see Figure 10), which as 28 layers in total, and found that modifying layers 4 through 26 yields the best cross-task performance, considering the number of skipped layers. This aligns with the common observation that early and final layers are more sensitive and have a disproportionate impact on final logits.

**Division between base and dynamic neurons**. We tested values of $\delta$ from 25% to 35% in small increments (Figure 11) using the Llama3.2-3B model. The accuracy remains stable across this range, with $\delta = 30\%$ slightly outperforming the alternatives, making it a reasonable and robust choice.

## B    LLM PROFILER COST

Table 5 summarizes the offline profiling cost across all models. All results were obtained on an NVIDIA L40S GPU using a batch size of 16. For the 3B-parameter models Qwen2.5-3B and Llama3.2-3B, the profiler processes roughly 44M tokens in 5.0–5.5 hours, achieving throughputs of 2.2k–2.4k tokens/s with peak GPU memory usage between 16 GB and 21.8 GB. For Llama3.2-1B on the same 43.8M-token corpus, profiling completes in 2.69 hours at 4.5k tokens/s with a peak of 10.8 GB GPU memory, while using a smaller 8.8M-token corpus reduces the wall-clock time to 0.56 hours at 4.3k tokens/s and 7.6 GB peak memory.

