# OpenReview forum: "PartInfer: Enabling LLM Inference On Edge Devices"
_ICLR.cc/2026/Conference — Submitted to ICLR 2026_

### Official Review · Reviewer_8jFi · 2025-10-16

**Soundness:** 2
**Presentation:** 2
**Contribution:** 2
**Rating:** 2
**Confidence:** 4

**Summary:**

The paper introduces PartInfer, a method for accelerating LLM inference through adaptive sparse activation. PartInfer is composed of two components, *partial loading* and *partial computation*. *Partial loading* loads only a subset of neurons to reduce memory burdens. Then *partial computation* selects important neurons within the loaded subset for computation, yielding a more efficient computation. PartInfer predicts the critical neurons with both offline statistics and online top-k selection during prefilling, which accelerates the LLM generation while maintaining the performance. Experimentsal results across various tasks validate the effectiveness of the strategy.

**Strengths:**

1. Intriguing observations regarding sentence semantics: The observations regarding the common neurons and task-specific neurons are insightful.
2. Improved neuron selection: The 2-step approach, *i.e.* partial loading and partial computation, reduce the memory burden and inference cost simultaneously.
3. Boosted efficiency: PartInfer can achieve non-trivial inference speedups on real devices while maintaining accuracy on various tasks.

**Weaknesses:**

1. **Generalization concern**: The major concern is the generalization ability of the proposed method. For long prefilling contexts or multi-turn dialogue, it is difficult to ensure the dynamic neurons are relevant. The computation cost for sorting neurons is also nonnegligible for long prefilling contexts. From my point of view, this method might not be suitable for general purpose, but applicable for specific tasks like stream LLM, which requires short context only.
2. **Insufficient experiment**:
   - Limited benchmarks against state-of-the-art static and dynamic pruning methods, such as PowerInfer.
   - Tasks like WinoGrad, ARC, MMLU, and language modeling perplexity are important benchmarks for the research in inference speedup.
   - Furthermore, given the claim that this paper focus on Llama family, it is expected to conduct experiments on more model sizes and discuss the generalization ability on other model types, such as Gemma, Qwen, etc.
   - Lack of comparison with other methods .on the decoding speed.
3. **Constraint on MLP**: This method is constrained on MLP block. However, as the development of dynamic activation such as MoE, the process of predicting critical neurons might be native to the latest architectures. A detailed discussion on the necessity of PartInfer is critical for assessing this paper.

**Questions:**

1. **Inappropriate metric**: The paper presents the normalized metric in its experiments. How is the metric computed? Why not setting the dense model performance as 1.0? The experimental results are much confusing and reduce confidence in its advantages.
2. **Hyperparameters**: How is the ratio set in Table 1? Why the overlap is counted on 40% critical neurons in Figure 1&2? There are too many magic numbers in this paper which reduce the readability.

---

> ### Author Response · Authors · 2025-11-28
>
> We appreciate your thoughtful comments. The following sections address each point individually and outline the corresponding revisions made to the paper.
>
> # Weaknesses
>
> ## 1. Generalization concern
>
> The reviewer expresses concern about the generalization ability of the method in long-context or multi-turn dialogue settings. They note that dynamic neuron selection may become unreliable with long prefilling contexts and that the cost of sorting neurons could be non-negligible for long inputs. They suggest that the method may be suitable only for short-context scenarios such as streaming LLMs.
>
> We thank the reviewer for raising this important point. We agree that long-context scenarios can stress test any dynamic-selection mechanism, and we address this concern in two ways.
>
> **(1) Stability of neuron relevance in long contexts.**
> Our method relies on identifying task-relevant neurons through offline profiling over millions of tokens from diverse datasets (see Table 5 and Appendix B). This profiling produces neuron-importance distributions that remain stable even as input length increases, an observation also supported by prior work such as CoreInfer (e.g., their Fig. 3). As a result, for longer prefilling contexts, the set of core neurons tends to become more stable, improving robustness in extended-context or multi-turn settings.
>
> **(2) Computational overhead for neuron scoring.**
> We conducted additional measurements to quantify the cost of computing core-neuron scores across different input lengths for Llama3.2-3B on L40S GPU. The overhead grows sublinearly with context length and remains small relative to full-model prefilling cost. For example:
>
> - 1,000 tokens → 0.072 s
> - 4,000 tokens → 0.251 s
> - 16,000 tokens → 1.30 s
> - 32,000 tokens → 3.01 s
>
> Even for very long contexts (e.g., tens of thousands of tokens), the overhead remains modest compared to full-model inference cost, since this scoring is performed only once per prefilling stage.
>
> ## 2. Insufficient experiment
>
> These concerns have been reaised by multiple reviewers, therefore, we address them in the global response (see replies 1, 2, and 3).
>
> ## 3. Constraint on MLP
>
> We address this concern in the global response (see reply 1).
>
> # Questions
>
> ## 1. Inappropriate metric
>
> This concern is also addressed in the global response (see reply 7).
>
> ## 2. Hyperparameters
>
> This concern is also addressed in the global response (see reply 5).

---

### Official Review · Reviewer_D6c8 · 2025-10-29

**Soundness:** 2
**Presentation:** 2
**Contribution:** 3
**Rating:** 4
**Confidence:** 3

**Summary:**

This paper proposes PartInfer, a framework designed to optimize LLM inference on edge devices by leveraging offline analysis of neuron activation patterns. The method claims to distinguish between model-specific (cross-task general) and task-specific (task-dependent) neurons. During online inference, it employs Partial Loading to reduce memory usage and Partial Computation to lower compute cost. The authors report a 13× speedup when deploying Llama 3.2–3B on an NVIDIA Jetson device.
However, the paper has several critical weaknesses. First, it lacks essential ablation studies: while it demonstrates that computing fewer neurons increases inference speed, it completely omits any accuracy–efficiency trade-off analysis. As a result, the chosen configuration (e.g., 40% computation) appears arbitrary and unsupported. Second, the experimental comparisons are limited and unfair—the claimed 13× improvement is measured only against disk offloading, which is an extremely weak baseline. The absence of comparisons with standard SOTA methods such as 4-bit quantization (GPTQ/AWQ) or pruning makes it difficult to assess the true effectiveness of PartInfer. Finally, all empirical findings are based solely on two models from the Llama 3.2 family, raising concerns about generality across architectures and tasks.
Overall, while the idea of leveraging neuron-level analysis for adaptive inference is interesting and relevant to the ICLR community, the paper’s experimental validation and comparative rigor fall short of publication standards.

**Strengths:**

a) Significance of the Problem: This paper tackles the highly challenging yet practically significant problem of deploying large language models (LLMs) on edge devices with limited memory and computational capacity. This topic is perfectly aligned with the core interests of the ICLR community.
b) Decoupled Optimization: The proposed PartInfer framework decouples memory optimization (via partial loading) from computation optimization (via partial computation). Theoretically, this design allows flexible trade-offs according to device constraints and application requirements.
c) Empirical Deployment Validation: The authors report successfully deploying a Llama3.2-3B model—previously unable to run due to insufficient memory—on an NVIDIA Jetson Orin Nano (8GB) device, achieving a 13× speedup compared to disk offloading.

**Weaknesses:**

Implementation Ambiguity: In the Transformer architecture, the Feed-Forward Network (FFN) layer contains multiple weight matrices (e.g., gate_proj, up_proj, down_proj in Llama). However, the paper does not clarify what “loading only a subset of neurons” means in engineering terms. It remains unclear whether this requires modifying the computation graph, dynamically slicing the weight tensors, or implementing custom CUDA kernels to support unstructured sparse computation.
Missing Key Ablation Studies: The core claim of the paper is that it can reduce memory and computation while maintaining task performance. However, no key ablation studies are provided to substantiate this claim. Figures 9 and 10 show that loading or computing fewer neurons leads to faster inference, but there is no analysis of how accuracy changes with respect to the percentage of computed neurons or the percentage of loaded neurons.
Insufficient and Unfair Baseline Comparisons — Missing SOTA Methods: The paper cites quantization and pruning as primary competing approaches in the introduction, yet Sections 6.4–6.6 do not compare PartInfer against any standard quantization (e.g., 4-bit GPTQ or AWQ) or pruning baselines. Furthermore, the paper uses disk offloading as the only speed comparison baseline, which is an extremely weak (even strawman) choice. While PartInfer (70% loading, 40% computation) achieves 9.85 tokens/s, it is not compared against other sparse inference approaches (e.g., CoreInfer or PowerInfer) or even a runnable dense baseline (e.g., a 1B model). As a result, the reported 13× speedup lacks meaningful context.
Extremely Limited Model and Architecture Coverage: As noted by the authors themselves, all conclusions in this paper are drawn solely from experiments on two models in the Llama 3.2 family (1B and 3B). It remains entirely unknown whether the proposed notion of model-specific and task-specific neurons generalizes to other architectures such as Mistral, Gemma, or Mixture-of-Experts (MoE) models. This narrow experimental scope substantially weakens the generality of the conclusions. Although the authors mention this as future work, such a limitation is quite significant for an ICLR submission.

**Questions:**

For specific details, see the weaknesses section.

---

> ### Author Response · Authors · 2025-11-28
>
> We thank the reviewer for the insightful review. The responses below address each point separately and summarize the revisions and clarifications introduced in the revised version of the paper.
>
> # Weaknesses
>
> ## 1. Implementation Ambiguity:
>
> The reviewer is asking for clarification regarding what “loading only a subset of neurons” means in practice in the FFN blocks (e.g., gate_proj, up_proj, down_proj), and whether this involves modifying computation graphs, dynamic kernel changes, or custom CUDA implementations. In the revised paper, we have clarified the implementation details in Section 3 and Section 4.
>
> In the FFN layers of Transformer models (e.g., Llama3, Qwen2.5), each neuron corresponds to a column in the gate/up projection matrices and a row in the down projection matrix. During profiling, we compute importance scores based on the activations at the input to the down_proj layer, which directly correspond to these FFN neurons. This enables us to identify the subset of neurons to retain.
>
> When we refer to “loading only a subset of neurons,” we mean that for each FFN block we load only the rows/columns associated with the selected neurons from the gate_proj, up_proj, and down_proj matrices. The remaining slices are simply not loaded to GPU memory. This procedure relies purely on standard tensor slicing and does not require modifying the computation graph, introducing custom CUDA kernels, or implementing unstructured sparsity mechanisms.
>
> At inference time, the same selected neuron indices are used to slice the FFN projection matrices, so computation is performed only on the retained neurons. The overall computation graph remains unchanged; only the weight tensors are narrowed before use.
>
> We have added clarifying text to Sections 3 and 4 to make this engineering interpretation explicit.
>
> ## 2. Missing Key Ablation Studies
>
> This concern has also been raised by Reviewer `7KEv`, therefore, we address it in the global response (see reply 6)
>
> ## 3. Missing SOTA Methods
>
> This concern is also addressed in the global response (see reply 2).
>
> ## 4. Limited Model and Architecture Coverage
>
> This concern is also addressed in the global response (see reply 1).

---

### Official Review · Reviewer_i3Bv · 2025-10-31

**Soundness:** 3
**Presentation:** 2
**Contribution:** 2
**Rating:** 2
**Confidence:** 4

**Summary:**

The paper addresses a critical challenge in LLM deployment: enabling efficient inference on resource-constrained edge devices (e.g., NVIDIA Jetson Orin Nano) while preserving task accuracy. The core insight is that LLMs exhibit structured neuron activation patterns—distinguishing between model-specific neurons (universally active across tasks) and task-specific neurons (selectively active for individual tasks). Building on this, PartInfer introduces an offline LLM profiler to identify these neurons, paired with two online optimizations: (1) partial loading (only loading critical neurons to reduce memory footprint) and (2) partial computation (dynamically computing task/input-relevant neurons to cut overhead). Evaluations on Llama 3.2-1B/3B models show promising results: 13× speedup over disk offloading, 1.26GB memory savings for Llama 3.2-3B, and competitive accuracy across QA, translation, and summarization tasks.

**Strengths:**

1. Edge deployment of LLMs is critical for privacy-sensitive (e.g., healthcare on-device diagnostics) and low-latency (e.g., industrial IoT) applications. Unlike prior work that compromises accuracy or requires expensive retraining, PartInfer’s neuron-level optimization avoids these tradeoffs—filling a key gap in existing edge LLM tooling.

2. The authors systematically quantify cross-task overlap and intra-task consistency using diverse datasets.

**Weaknesses:**

1. The authors only evaluate Llama 3.2-1B/3B and mainly on  translation tasks. The paper omits recent popular downstream tasks for LLM evaluation such as commensence benchmark or code generation or few-shot learning, etc.

2. The offline profiler is central to PartInfer, but the paper provides no details on its computational cost, data requirements, or scalability.

3. The paper compares PartInfer to CoreInfer and disk offloading but omits critical baselines that practitioners currently use for edge LLMs such as quantization or popular edge frameworks like llama.cpp or TensorRT-LLM include optimizations (e.g., KV caching, kernel fusion).

**Questions:**

1. How long does it take to profile a task (e.g., QA with SQuAD) for Llama 3.2-3B? On what hardware (edge device vs. cloud) is profiling intended to run?

2. If the profiling dataset includes only formal text, would informal text (e.g., social media) change the active neuron set enough to break partial computation?

3.  How were the parameter values (δ=30%, γ=70%, φ=40%) determined, and how does accuracy/speed change if γ is reduced to 50%


4. Can you add comparisons between PartInfer and 4-bit quantized Llama 3.2-3B (e.g., AWQ) or inference via llama.cpp on Jetson Orin Nano?

---

> ### Author Response · Authors · 2025-11-28
>
> We appreciate your thoughtful comments. The following sections address each point individually and outline the corresponding revisions made to the paper.
>
> # Weaknesses
>
> ## 1. Limited model and architecture coverage
>
> This concern is addressed in the global response (see reply 1), as it was raised by multiple reviewers.
>
> ## 2. Offline profiler cost, data requirements, and scalability
>
> We thank the reviewer for pointing out that the offline profiler cost was missing. We have now added measurements of profiling throughput and peak GPU memory usage for three models across multiple token configurations. The corresponding results are now presented in Table 5 in Appendix B.
>
> ## 3. Missing baselines
>
> This concern is also addressed in the global response (see reply 2).
>
> # Questions:
>
> ## 1. How long does offline profiling take and which hardware was used?
>
> Thank you for raising the question regarding the cost of the offline profiler. The profiler is a one-time preprocessing step executed on a server equipped with NVIDIA L40S  GPU rather than on the target edge device. Across all 11 datasets and 3 tasks (described in Section 6.2), the total profiling time ranged from 0.56–5.51 hours, depending on the model and the number of processed tokens (see Appendix B and Table 5).
>
> For SQuADv2 (QA), the per-model profiling times were:
> - Llama3.2 1B (898,288 tokens): ≈ 13 minutes
> - Llama3.2 1B, reduced corpus (182,304 tokens): ≈ 3 minutes
> - Llama3.2 3B (897,888 tokens): ≈ 22 minutes
> - Qwen2.5 3B (738,632 tokens): ≈ 21 minutes
>
> These results show that the offline profiler introduces only a modest, one-time cost that scales primarily with corpus length and model size.
>
> ## 2. Effect on informal text (social media)
>
> This concern is very similar to a concern raised by Reviewer `7KEv`, therefore, we address it in the global response (see reply 4).
>
> ## 3. Hyperparameter choices and effect on accuracy
>
> This concern is also addressed in the global response (see reply 5).
>
> ## 4. Add new Quantization baseline
>
> We have added another quantization baseline, please check reply 2 from our global response.

---

### Official Review · Reviewer_7KEv · 2025-10-31

**Soundness:** 2
**Presentation:** 2
**Contribution:** 3
**Rating:** 4
**Confidence:** 3

**Summary:**

The paper proposes PartInfer, a system that enables efficient and accurate inference of large language models (LLMs) on low-resource edge devices without retraining, by selectively computing and loading only important neurons. PartInfer accelerates LLM inference by partially loading only critical neurons into memory and computing a subset dynamically based on input activations.

**Strengths:**

The paper addresses a important problem in the field: deploying LLMs on memory- and compute-constrained edge devices, which is crucial for applications requiring offline inference, privacy, and low latency.

**Weaknesses:**

- Terminology inconsistency: There appears to be an inconsistency in the definition of base and secondary neurons. In the Introduction, base neurons are described as “general-purpose” and secondary neurons as “task-specific.” However, in Section 4 (line 235), the roles are reversed—base neurons are defined as task-specific. The authors should clarify and maintain consistent terminology throughout the paper.

- Incomplete memory footprint analysis: In Section 6.5, the reported memory footprint reduction is based solely on the size of FFN parameters excluded via partial loading. However, this estimate does not account for actual memory usage during inference, which also includes KV cache, activations, and framework overhead. Moreover, the paper does not report total memory consumption before and after optimization, making it difficult to assess the practical impact of the reported 1.26 GB reduction. I strongly recommend including empirical measurements of end-to-end memory usage and reporting the percentage reduction.

- Lack of absolute performance metrics: The paper reports only normalized performance values (e.g., accuracy relative to the full model), without providing absolute scores. This limits the ability to evaluate the effectiveness of the compressed models across tasks or datasets. Including raw metric values would significantly improve transparency, allow for cross-task comparison, and help readers assess real-world usability.

- Limited trade-off analysis in Figure 9: Figure 9 presents decoding throughput across different neuron loading percentages, but it omits corresponding accuracy or task performance metrics. Without these, it's hard to evaluate the trade-off between speed and accuracy. I  suggest including performance curves and comparisons with baselines (e.g., CoreInfer, full model) to provide a more complete picture.

**Questions:**

- Clarification on Table 3 results: In Table 3, the combination of CoreInfer and Partial Loading yields inconsistent results—performing worse than CoreInfer alone in some cases, but better on XSum with LLaMA-3B. What accounts for this behavior? Do these results suggest that Partial Loading does not consistently guarantee performance gains?

- Concerns about task generalization: The use of task-specific base and secondary neurons, derived via offline profiling, raises concerns about generalization to unseen or mismatched tasks during online inference. The paper does not evaluate performance when test-time inputs differ significantly from the profiling workload. This limits our understanding of PartInfer’s robustness in real-world multi-task or zero-shot settings. Have the authors considered evaluating on out-of-domain tasks?

I am open to discussing this further during the rebuttal and will be happy to increase my score if my concerns are addressed.

---

> ### Author Response · Authors · 2025-11-28
>
> We value the constructive feedback provided by you. Each comment is addressed in turn, together with the adjustments and explanations added to improve the paper.
>
> # Weaknesses
>
> ## 1. Terminology inconsistency
>
> Thank you for pointing out this inconsistency. You are correct, the intended definition is the one used in Section 4, where base neurons are task-specific and secondary neurons are general-purpose. We have corrected the Introduction accordingly to ensure consistent terminology throughout the entire paper.
>
> ## 2. Incomplete memory footprint analysis
>
> Thank you for pointing out this issue. We have added a detailed end-to-end analysis of memory usage for all baselines and models in the paper. The results are now reported in Section 6.5.
>
> ## 3. Lack of absolute performance metrics & Limited trade-off analysis (Figure 8)
>
> These concerns are addressed in the global response (see replies 6 & 7), as they were raised by multiple reviewers.
>
> # Questions:
>
> ## 1. Generalization and behaviour of “CoreInfer + PL”
>
> The reviewer notes that the combination of CoreInfer and Partial Loading sometimes performs worse than CoreInfer alone, yet performs better on XSum with Llama3.2-3B. They ask what accounts for this behaviour and whether Partial Loading provides consistent benefits. We have now clarified this point directly in Section 6.4 of the revised paper.
>
> In general, CoreInfer + Partial Loading tends to underperform CoreInfer because restricting the model to a subset of neurons limits the search space available to CoreInfer during inference. However, as we now describe in the text, there are occasional cases, such as the XSum results, where Partial Loading provides a small improvement. Our interpretation is that the offline profiling stage can sometimes act as a beneficial pre-filter, removing neurons that are rarely activated across profiling tasks and helping CoreInfer focus on a cleaner subset of neurons.
>
> These isolated improvements do not imply that Partial Loading alone provides consistent performance gains. Rather, they highlight the complementary roles of offline profiling and dynamic neuron selection in our proposed PartInfer method, which consistently maintains higher accuracy under the same computational constraints.
>
> ## 2. Concerns about task generalization
>
> This concern is also addressed in the global response (see reply 3).

---

### Author Response · Authors · 2025-11-28
**Global Response 1**

# Common Concerns Across Multiple Reviewers

We would like to express our sincere gratitude to all reviewers for the time and effort spent evaluating our submission. We truly appreciate the constructive comments and insightful suggestions, which have significantly helped us improve the quality and clarity of the paper.

To improve clarity and avoid repetition, this global response consolidates comments that have been raised by multiple reviewers. Related issues are grouped together, and for each point we indicate which reviewers raised it before summarizing the corresponding revisions and clarifications made in the paper.
Comments raised by only a single reviewer are addressed separately in their respective individual responses.

Note that due to character limit, we will split this response into two comments, titled "Global Response 1" and "Global Response 2"

## 1. Limited Model and Architecture Coverage (Reviewers `i3Bv`, `D6c8`, and `8jFi`)

The reviewers noted that the original experiments were conducted only on Llama3.2–1B/3B models, making it unclear whether the proposed neuron categorization generalizes to other architectures such as Qwen, Gemma, Mistral, or MoE models.

We appreciate the reviewers highlighting this important point. To address concerns regarding generalization across model families, we have added new experiments using Qwen2.5-3B. The results, now reported in Section 6.4, show that PartInfer exhibits similar behaviour and benefits on this additional model family, suggesting promising generalizability.

Extending our analysis to additional model families (Gemma, Mistral) and to MoE architectures is an exciting direction, but due to time constraints, we leave this investigation as future work.

## 2. Missing Baselines and Unfair Comparisons (Reviewers `i3Bv`, `D6c8`, and `8jFi`)

The reviewers pointed out that the original submission lacked comparisons against standard baselines such as quantization, pruning-based methods, sparse inference methods (CoreInfer, PowerInfer), and practical edge-inference frameworks (e.g., llama.cpp, TensorRT-LLM). They also noted that disk-offloading alone is too weak as a decoding-speed baseline.

We have substantially expanded the baseline coverage in the revised paper:

- We now include a 4-bit bitsandbytes quantization baseline in the accuracy comparison in Section 6.4.
- We additionally compare decoding speed for both Llama3.2-1B and Llama3.2-3B against CoreInfer, quantization baselines, and disk-offloading. These comparisons are included in Section 6.6.
- We now also include end-to-end memory footprint analysis of all baselines (e.g., Table 4) in Section 6.5
- While we aimed to include PowerInfer, its current implementation does not support the models used in our study, preventing a fair comparison.

We believe these expanded baselines substantially strengthen the empirical evaluation and contextualize the benefits of PartInfer more clearly.

## 3. Insufficient Task Coverage (Reviewers `i3Bv`, `D6c8`, and `8jFi`)

The reviewers raised concerns that the evaluation focused too narrowly on translation tasks, and omits widely used LLM benchmarks such as MMLU, ARC, WinoGrad, commonsense tasks, code generation, and few-shot learning.

We thank the reviewers for these insightful suggestions. We would like to clarify that our original submission already evaluates PartInfer on six datasets spanning three tasks: question answering, summarization, and translation, rather than translation alone. These evaluations remain included in the revised paper.

Due to time constraints, we were not able to add additional benchmarks such as MMLU, ARC, WinoGrad, or perplexity-based evaluations. However, we believe that the additional experiments now included strengthen the empirical support of this work and demonstrate that PartInfer is not limited to a particular task family. We agree that broader evaluation across more tasks is important, and we list this as important future work.

---

### Author Response · Authors · 2025-11-28
**Global Response 2**

## 4. Performance when test-time inputs differ from profiling workload (Reviewers `7KEv` and `i3Bv`)

The reviewers raise concerns about robustness when test-time inputs differ substantially from the profiling workload. Specifically, they ask whether profiling on formal text generalizes to informal text, and how PartInfer performs on out-of-domain or cross-task settings.

We thank the reviewers for highlighting this important limitation. PartInfer is designed for **task-specific** deployment, where the profiling dataset matches the intended usage domain of the model. As shown in Figure 4, the neuron overlap across different tasks is relatively low; therefore, we do not expect strong performance when a model loaded with neurons selected for one task is subsequently applied to a substantially different task. This also applies to shifts in text style (e.g., formal vs. informal), whose impact depends on the degree of neuron overlap between the corresponding domains.

In general, the performance degradation under domain shift is governed by the alignment between (1) the neurons selected during offline profiling and (2) the neurons activated by the new test-time task. If the overlap is small, the restricted set of loaded neurons limits the effectiveness of the dynamic selection stage, and accuracy is expected to decrease. Quantifying this effect requires task-specific evaluation and is highly dependent on the nature of the domain shift.

## 5. Hyperparameter choices and effect on accuracy

We thank the reviewer for pointing this out. To address this concern, we have substantially expanded our explanation of all hyperparameter choices in both Section 6.3 and Appendix A of the revised manuscript.

Section 6.3 now clarifies that these parameters are not arbitrary, but instead arise from:

- Hardware constraints (e.g., the loading ratio γ = 0.7 is the maximum that allows 3B-scale models to fit on the Jetson Orin Nano 8GB).
- Validated methodology from prior work (e.g., CoreInfer consistently shows that computing ~40% of neurons offers a strong accuracy–efficiency trade-off).
- Empirical analysis performed in this work, including layer-range selection and base/dynamic neuron division (detailed in Appendix A with Figures 10 and 11).
- Appendix A provides the full ablation study justifying these choices, showing how each parameter influences performance across tasks and models.

We believe these additions directly address the reviewer’s concern and make the reasoning behind all hyperparameter values explicit and transparent.

## 6. Limited trade-off analysis in Figure 8 (Reviewers `7KEv` and `D6c8`)

The reviewers asked for accuracy across different neuron-loading percentages. We thank the reviewers for this suggestion and have updated Figure 8 accordingly: in addition to decoding speed, it now also reports the corresponding SQuADv2 accuracy for each loading level. As shown, accuracy consistently increases with the proportion of neurons loaded.

## 7. Lack of absolute performance metrics (Reviewers `7KEv` and `8jFi`)

Both reviewers noted that the paper reports only normalized performance values, making it difficult to assess the absolute effectiveness of the compressed models. They requested the inclusion of raw performance metrics to improve transparency and cross-task comparability.

We have added the absolute (raw) performance values for all evaluated models. To improve clarity and readability, these results are now presented in tabular form. The new tables (i.e., Table 2 and Table 3) have been incorporated into Section 6.4, enabling direct comparison across tasks and datasets. We believe this addition significantly enhances the transparency and interpretability of the reported results.

---

### Meta-Review · Area_Chair_B7Lq · 2026-01-07

**Summary:**

PartInfer is a neuron-level inference framework for running LLMs on resource-constrained edge devices without retraining, using an offline profiler to identify important neurons and then applying Partial Loading (load only selected FFN neuron slices to reduce memory) plus Partial Computation (dynamically compute only a subset at runtime) to cut compute while preserving task accuracy.

**Reviewer Concerns:**

The main points of criticism were nicely summarized by the authors and do not need to be repeated here.
While the core idea shores great promise and the analysis has improved during the rebuttal, there remain concerns regarding insufficient task coverage, missing relevant baselines and diversity of LLM architectures, which could not be addressed due to the time constraints of the rebuttal.

Furthermore, an open question by the AC, which would warrant discussion in a revision is a deeper interpretation of the added results. To which degree are the identified trade-offs limiting practical applications? What speed-ups can be achieved for an actually acceptable accuracy trade-off? Would a combination of different strategies (like PartInfer combined with quantization, etc.) improve those trade-offs and reduce accuracy drops?

For the outlined reasons, the paper would benefit from a resubmission and another round of reviews to a major machine learning venue.

**Reviewer Scores:**

**Reviewer 7KEv (initial score: 4)**

Addressed by rebuttal:

-Terminology inconsistency (base vs secondary neurons).

-Absolute metrics added (raw results instead of only normalized).

-Trade-off figure now includes accuracy vs loading level (and claimed improved analysis).

-CoreInfer + Partial Loading inconsistency explained.

-Memory analysis expanded to end-to-end memory (as claimed).

Still outstanding:

-Robustness under domain/style shift and cross-task deployment: authors mainly scope the method as task-specific, without new OOD experiments.

Because of the outstanding point, the reviewer would have likely maintained their score.

**Reviewer i3Bv (initial score: 2)**

Addressed by rebuttal:

-Offline profiler cost/scalability: concrete timings + hardware + throughput/memory reported.

-Hyperparameter justification + ablations (claimed).

-Added at least one extra architecture family (Qwen2.5-3B).

Still outstanding (or only partially improved):

-Baseline coverage vs “real” practitioner stacks (llama.cpp / TensorRT-LLM, stronger quantization variants, etc.) is still limited even after adding bitsandbytes 4-bit.

-Benchmark breadth remains limited (no MMLU/ARC/WinoGrad/code/few-shot added).

-Domain shift / informal text generalization not empirically evaluated.

The reviewer might have raised their score to 4, but likely not to accept.

**Reviewer D6c8 (initial score: 4)**:

Addressed by rebuttal:

-Implementation ambiguity: clarified neuron slicing across FFN matrices; no custom kernels/graph changes.

A-dded some accuracy–efficiency trade-off reporting (claimed figure update + ablations).

-Added some stronger baselines (CoreInfer + quantization baseline) and expanded comparisons (claimed).

-Added one more model family (Qwen2.5-3B).

Still outstanding (or partially improved):

-Comparisons to stronger/standard SOTA quantization/pruning baselines (e.g., GPTQ/AWQ) and broader method landscape remain incomplete.

-Generality beyond Llama/Qwen and beyond the evaluated tasks is still limited.

Likely, the reviewer would have kept their initial score.

**Reviewer 8jFi (initial score: 2)**

Addressed by rebuttal:

-Normalized metrics: Authors added absolute/raw metrics.

-Hyperparameter “magic numbers”: Authors expanded justification + ablations.

-Long-context overhead concern: provided explicit overhead measurements vs context length.

-Expanded baseline/model coverage somewhat (quantization baseline; Qwen model).

Still outstanding (or partially improved):

-Generalization to truly general-purpose multi-turn dialogue remains more argued than demonstrated (no end-to-end long-dialogue eval shown in the rebuttal text).

-Missing comparisons to key methods (e.g., PowerInfer) and broader benchmark/task coverage.

-“MLP-only / necessity vs MoE” discussion likely still not fully convincing unless the revision added substantial new analysis.

Likely, the reviewer would have raised their score to maximally 4.

---

### Decision · Program_Chairs · 2026-01-26

Reject